# Does the Coronoid Always Need to Be Fixed in Terrible Triad Injuries of the Elbow? Mid-Term Postoperative Outcomes Following a Standardized Protocol

**DOI:** 10.3390/jcm9113500

**Published:** 2020-10-29

**Authors:** Beom-Soo Kim, Du-Han Kim, Seok-Ho Byun, Chul-Hyun Cho

**Affiliations:** Department of Orthopedic Surgery, Keimyung University Dongsan Hospital, Keimyung University School of Medicine, 1035 Dalgubeol-daero, Dalseo-gu, Daegu 42601, Korea; kbs090216@gmail.com (B.-S.K.); osmdkdh@gmail.com (D.-H.K.); eddy2149@dsmc.or.kr (S.-H.B.)

**Keywords:** elbow, coronoid, terrible triad injuries, clinical protocol, outcomes

## Abstract

The purpose of this study was to investigate mid-term outcomes and complications after operative treatment according to a standardized protocol for terrible triad injuries. Twenty-four patients that were treated by a single surgeon with a standardized surgical protocol were retrospectively reviewed. After the complete reconstruction of radial head and/or lateral collateral ligament (LCL) complex through a lateral approach, coronoid process, and/or medial collateral ligament (MCL) complex through a medial approach were fixed if the elbow is unstable. For coronoid fractures, only type III were fixed in four cases (16.7%). Twenty-two LCL (91.7%) and five MCL (20.8%) complexes were repaired. At the final follow-up, the mean MEPS and Quick-DASH score were 91.5 and 17.3, respectively. There was no recurrent instability after operation in all cases. This study revealed that operative treatment that was based on our standardized protocol for terrible triad injuries yielded satisfactory mid-term clinical and radiographic outcomes without any recurrent instability. These results suggest that Type I and II coronoid fractures in terrible triad injuries do not need to be fixed if the radial head and ligamentous complex are completely reconstructed.

## 1. Introduction

Although the combination of radial head and coronoid process fractures with posterior elbow dislocation is a common injury, its treatment has been traditionally associated with unsatisfactory outcomes and high complication rates [1,2,3,4,5]. For this reason, it has been called the “terrible triad injury” of the elbow [3] Terrible triad injuries can lead to pain, recurrent instability, stiffness, and posttraumatic arthritis if not properly treated [6]. The goal of treatment for these injuries is to restore the bony and ligamentous structures and provide sufficient stability for early range of motion (ROM) [5]. Recently, clinical and radiographic outcomes after operative treatment have improved with an improved understanding of elbow biomechanics and surgical technique with new implants [5,7,8]. However, until now, there is no clear consensus regarding surgical approach, implant choice, and essential structures that should be restored for sufficient elbow stability.

The operative treatment of the terrible triad injuries includes radial head fixation or replacement, coronoid fixation, and lateral collateral ligament (LCL) complex repair [8,9,10,11,12,13,14,15,16]. If the elbow is unstable, then medial collateral ligament (MCL) complex repair or hinged external fixation are sometimes required [14,15,16]. The most debatable issues for the treatment of these injuries are: (i) which structures to repair and (ii) what steps should be taken in order to reconstruct fractures and ligamentous injuries. Many surgeons have proposed their own treatment algorithms that they believe to contribute most substantially to elbow stability [7,8,9,12,13,14,16,17]. While a literature review revealed consensus in the requirement for the reconstruction of the radial head fracture and LCL complex tear, opinions vary as to whether coronoid process fractures require repair [10]. Although most of the studies recommended the reconstruction of both radial head and coronoid process fractures in terrible triad injuries [7,8,9,12,14], recent cadaveric and clinical studies have reported that type I or II coronoid process fractures according to Regan–Morrey classification [18] can be stable if the radial head and ligamentous complexes are completely restored [6,19]. Papatheodorou et al. [6] reported on a series of 14 terrible triad injuries that were treated without fixation for type I or II coronoid fractures and achieved excellent results without any residual instability.

Based on the author’s concept and clinical experience, it was thought that coronoid process fractures do not always require fixation in order to provide sufficient elbow stability in terrible triad injuries. The purpose of this study was to investigate mid-term outcomes and complications after operative treatment according to a standardized protocol for terrible triad injuries. This study was conducted to test whether the hypothesis that type I and II coronoid fractures can achieve stability without fixation if the radial head and ligamentous complex are completely reconstructed.

## 2. Patients and Methods

This was a retrospective study of 27 patients with terrible triad injuries, who were treated by a senior surgeon (CHC) in a single fellowship-training hospital between 2008 and 2018 (Table 1). The inclusion criteria required patients to have radial head and coronoid process fractures with posterior elbow dislocation, as confirmed by plain radiographs or computed tomography (CT) scan at the time of the initial injury. The exclusion criteria were: (1) any history of previous elbow trauma or surgery, (2) the presence of elbow osteoarthritis before the injury, and (3) a follow-up period less than 24 months after operation. Twenty-seven patients met the inclusion criteria and were selected for this study. Three patients had loss of follow-up after surgery. Finally, 24 patients were included in this study and their cases retrospectively reviewed. This study was approved by our Institutional Review Board with the exemption of informed consent (IRB No. 202006058).

Mean patient age was 47.7 years (range: 19–77) and there were 12 men and 12 women. Right elbows were involved in 13 cases and left elbow in the remaining 11. The mechanism of injury included slip down (*n* = 11), fall from a height (*n* = 5), fall down (*n* = 5), sports injury (*n* = 2), and motor vehicle accident (*n* = 1). The radial head fractures were assessed according to Mason classification [20] and they were type III (*n* = 19), type II (*n* = 4), and type I (*n* = 1). Coronoid process fractures were assessed according to Regan–Morrey classification [18] and O’Driscoll classification [21]. According to the Regan–Morrey classification, there were type II (*n* = 12), type I (*n* = 8), and type III (*n* = 4). According to O’Driscoll classification, there were tip type (*n* = 12), anteromedial type (*n* = 8), and basal type (*n* = 4). Two cases had an open fracture. The mean time from initial trauma to operation was 5.3 days (range: 1–18 days).

### 2.1. Surgical Procedure

All of the patients were treated early with surgical reconstruction by a senior surgeon (CHC) with a standardized protocol (Figure 1). In all cases, a lateral approach was performed through the Kocher interval. After complete reconstruction of the radial head and/or LCL complex, elbow stability, including valgus stress test, was assessed under the image intensifier (Figure 2). If instability remained, then the coronoid process and/or MCL complex were fixed through a medial approach. In cases with type I or II coronoid fractures, torn MCL complexes were repaired without coronoid fragment fixation. All type III coronoid fractures were fixed. The internal fixation of the radial head was performed in eight cases (plate fixation = 4, headless screw fixation = 4), replacement in 15 cases (8 Evolve^TM^, Wright Medical Technology, Memphis, TN, USA; 3 Anatomic Radial Head System, Acumed, Hillsboro, OR, USA; 3 ExploR, Zimmer-Biomet, Warsaw, IN, USA; 1 RHS^TM^, Tornier, Montbonnot-Saint-Martin, France), and no fixation in one case with nondisplaced marginal fracture. Only Type III coronoid fractures were fixed in four cases (plate fixation = 3, screw fixation = 1). Twenty-two LCL and five MCL complexes were repaired using suture anchors (3.5 or 5.0 Twinfix Ti, Smith & Nephew, Fort Worth, TX, USA). After surgery, long arm splints were applied in 90° flexion and neutral position. Wearing hinged elbow brace, passive and active ROM exercise was started an average 7.4 days (range: 3–14 days) after surgery.

### 2.2. Outcome Assessment

The mean follow-up period was 65.0 months (range: 24–163). Final clinical outcomes were assessed while using the Mayo Elbow Performance Score (MEPS), the Quick Disabilities of the Arm, Shoulder and Hand (Quick-DASH) score, and active ROM of the elbow joint. Based on MEPS, the clinical outcomes were classified, as follows: excellent, ≥90 points; good, 75–89 points; fair, 60–74 points; and, poor, <60 points. Radiographic outcomes, including heterotopic ossification and posttraumatic arthritis, were assessed using serial plain radiographs. Heterotopic ossifications were assessed according Hastings and Graham classification [22]. Type II or III are regarded as clinically relevant heterotopic ossification. Posttraumatic arthritis was graded according to Bromberg–Morrey classification [23]: grade 0, normal elbow; grade 1, slight joint space narrowing with minimum osteophyte formation; grade 2, moderate joint space narrowing and moderate osteophyte formation; grade 3, severe degenerative changes and gross destruction of the joint. Complications were also evaluated.

### 2.3. Statistical Analysis

The SPSS statistical package (version 20.0; IBM, Armonk, NY, USA) was used for data analysis. In order to characterize any potential correlation between clinical outcomes and various parameters, such as age, sex, affected side, interval from initial trauma to operation, coronoid fixation, radial head fixation or replacement, and duration of immobilization, the chi-square test, Mann–Whitney U test, and Spearman correlation analysis were used [24,25]. Statistical significance was set at *p* < 0.05.

## 3. Results

Sixteen cases (66.7%) had radial head and/or LCL complex reconstruction through a lateral approach and sufficient elbow stability without any additional procedures. Out of the remaining eight cases with unstable elbows after lateral side reconstruction, four cases had MCL complex repair as an additional procedure, three cases had coronoid fixation, and one case had both coronoid fixation and MCL complex repair for sufficient elbow stability.

At the final follow-up, the mean MEPS and Quick-DASH score were 91.5 and 17.3, respectively. Seventeen patients (70.8%) had excellent outcomes, six (25.0%) good, and one (4.2%) fair. The mean ROM of the elbow was 131.7° of flexion, 3.5° of extension, 76.7° of pronation, and 79.6° of supination (Table 2) (Figure 3 and Figure 4). There were no significant correlations between clinical outcomes and parameters tested (i.e., age, sex, affected side, time from injury to operation, fixation of coronoid process, fixation or replacement of radial head, and duration of immobilization) (*p* > 0.05) (Table 3).

There was no cases of post-operative recurrent dislocation or subluxation. According to Hastings and Graham classification, nine cases had type I and 2 type IIA heterotopic ossification. Twelve cases had radiographic evidence of posttraumatic arthritis. According to Bromberg–Morrey classification, 12 cases had grade 0, 11 grade 1, and one grade 2 on final plain radiographs.

Six complications (25.0%), including transient ulnar neuropathy (*n* = 3), elbow stiffness with heterotopic ossification (*n* = 2), and olecranon fracture (*n* = 1), were observed. Three cases with transient ulnar neuropathy were completely recovered within eight weeks after primary operation.

Three cases (12.5%) needed reoperation. One patient (Case 10) with elbow stiffness with heterotopic ossification and progressive ulnar neuropathy underwent arthrolysis and anterior transposition of the ulnar nerve at six months after primary operation. The patient achieved excellent clinical outcomes at the final follow-up. One patient (Case 19) with elbow stiffness with heterotopic ossification underwent arthrolysis and in-situ decompression of the ulnar nerve at 5 months after primary operation. The patient had fair clinical outcomes at the final follow-up. One patient (Case 17) had olecranon fracture around the drill hole for passing a transosseous suture for triceps rupture as combine injury at seven weeks after primary operation and underwent plate fixation. The patient had excellent clinical outcome at the final follow-up.

## 4. Discussion

The present study was conducted in order to investigate whether fixation of type I and II coronoid fractures in terrible triad is required. According to the standardized surgical protocol described here, only type III coronoid fractures were fixed. Type I or II coronoid fractures were doing not underdo fixation after complete reconstruction of radial head and all ligamentous complex. Operative treatment for terrible triad injuries that were based on this standardized protocol yielded satisfactory mid-term clinical and radiographic outcomes. Additionally, there were no observed cases of post-operative recurrent instability. These results suggest that Type I and II coronoid fractures in terrible triad injuries do not require fixation if the radial head and ligamentous complex are completely reconstructed.

Most published treatment protocols of terrible triad injuries have advocated fixation or replacement of radial head fracture, fixation of coronoid fracture, and repair of the torn LCL complex [8,9,10,11,12,13,14,15,16]. MCL complex repair is required in cases with sustained elbow instability [14,15,18]. A hinged external fixator may be employed when sufficient stability of the elbow cannot be achieved after MCL complex repair [14,15,16]. Although these protocols have been shown to be generally effective, several studies reported high complication rates and fair clinical outcomes [7,9,12].

The most debatable issue for the treatment of terrible triad injuries is whether coronoid process fracture should be fixed. Because the coronoid process serves as a buttress for preventing posterior subluxation or dislocation, numerous studies have advocated for the fixation of all coronoid process fractures [7,8,9,12,13,14,16]. However, the management of coronoid process fractures may need to be differentiated based on fragment size and joint stability. A cadaveric study demonstrated that type I or II coronoid fractures are stable, unless the radial head is removed, whereas type III fractures are unstable, even with an intact radial head and LCL complex [19]. Several clinical studies also reported that small or very comminuted fragmentation of the coronoid process could be neglected without any fixation [6,26,27]. Antoni et al. [26] reported that reattaching the anterior capsule in terrible triad injuries with fractures of the coronoid tip did not improve the final clinical and radiographic outcomes.

They concluded that elbow stability can be achieved without coronoid fixation if a coronoid process fracture does not involve anteromedial facet or the fracture is less than 50% of the coronoid height [26]. Papatheodorou et al. [6] reported excellent results without any residual instability in a series of 14 terrible triad injuries with type I or II coronoid fractures that were treated without fixation. They concluded that type I or II coronoid fractures can be effectively treated without fixation when intraoperative elbow stability is achieved through functional ROM after reconstruction for a radial head fracture and LCL complex injury [6].

Another debatable issue is whether the MCL complex should be repaired. Chen et al. [10] reported that MCL complex repair can compensate for small coronoid deficiency. Beingessner et al. [27] reported that suture fixation of type I coronoid fractures has little effect on elbow stability. They suggested that MCL complex repair, rather than coronoid fixation, should be considered if the elbow remains unstable after the reconstruction of the radial head and LCL complex [27,28].

In the present study, our standardized operative protocol for terrible triad injuries is consistent with the concepts and opinions that were suggested by the above-mentioned studies [6,10,26,27]. Sixteen (66.7%) out of 24 cases were treated successfully with the reconstruction of the radial head and LCL complex through a lateral-only approach. Another four cases (16.7%) with residual instability after lateral side reconstruction were treated successfully with additional MCL complex repair in the absence of coronoid fixation through a medial approach. These results demonstrate that Type I and II coronoid fractures in terrible triad injuries do not require fixation if the radial head and ligamentous complex are completely reconstructed. The remaining four cases with type III coronoid fractures were treated with coronoid fixation and/or MCL complex repair after lateral side reconstruction. Numerous studies recommended a hinged external fixation in order to restore residual instability in some complex cases [3,5,10,14,15,29]. However, a hinged external fixation was not required, because no cases of recurrent instability after primary operation occurred.

Traditionally, a terrible triad injury has been considered to be a difficult-to-treat entity with unsatisfactory outcomes and high post-treatment complication rates [1,3]. However, recent studies reported improved post-operative clinical and radiographic outcomes as surgical techniques and implants have advanced [5,7,8]. The systematic review conducted by Chen et al. [29] analyzed the functional outcomes and complications of 312 patients with terrible triad injuries from 16 studies. The final MEPS ranged from 78 to 95 with a mean follow-up period of 25 to 30 months. Excellent outcomes were achieved in 39% of cases, while 43%, 15%, and 3% had good, fair, and poor outcomes, respectively.

The mean reoperation rate was 22.4% (70/312) (range: 0 to 54.5%). Most of these complications that require reoperations were related to hardware fixation problems, elbow stiffness, recurrent instability, and ulnar neuropathy. They concluded that functional outcomes after operation for terrible triad injuries are generally satisfactory but noted that complications are common. In the present study, final mean MEPS was 91.5. Seventeen patients (70.8%) had excellent outcomes, six (25.0%) good, and one (4.2%) fair. The mean ROM was 131.7° of flexion, 3.5° of extension, 76.7° of pronation, and 79.6° of supination. Three cases (12.5%) required reoperation. The results from our study are superior or similar to those of previously reported studies. Although the reasons why higher MEPS score and better ROMs with lower reoperation rates were achieved in this study remain unclear, we hypothesize that minimal fixation—just enough to provide elbow stability rather than complete restoration of all fractures—and ligamentous injuries led to improved outcomes. Additionally, the results from this and previous studies demonstrate that type I or II coronoid fractures can be effectively treated without any fixation if the radial head and ligamentous complex are completely reconstructed. The avoidance of unnecessary procedures can reduce surgical trauma and positively affect functional outcomes and reduce complications.

This study has several limitations. First, it is a retrospective study with a small number of patients. Second, clinical outcomes and complications according to coronoid fixation status were not directly compared because of the lack of a comparison group. Further long-term prospective studies are needed in order to compare outcomes, including complications according to coronoid fixation or treatment algorithm.

## 5. Conclusions

This study revealed that the standardized protocol for operative treatment of terrible triad injuries described herein yielded satisfactory mid-term clinical and radiographic outcomes without any recurrent instability. These results suggest that the fixation of Type I and II coronoid fractures in terrible triad injuries are not necessary if radial head and ligamentous complex are completely restored.

## Figures and Tables

**Figure 1 jcm-09-03500-f001:**
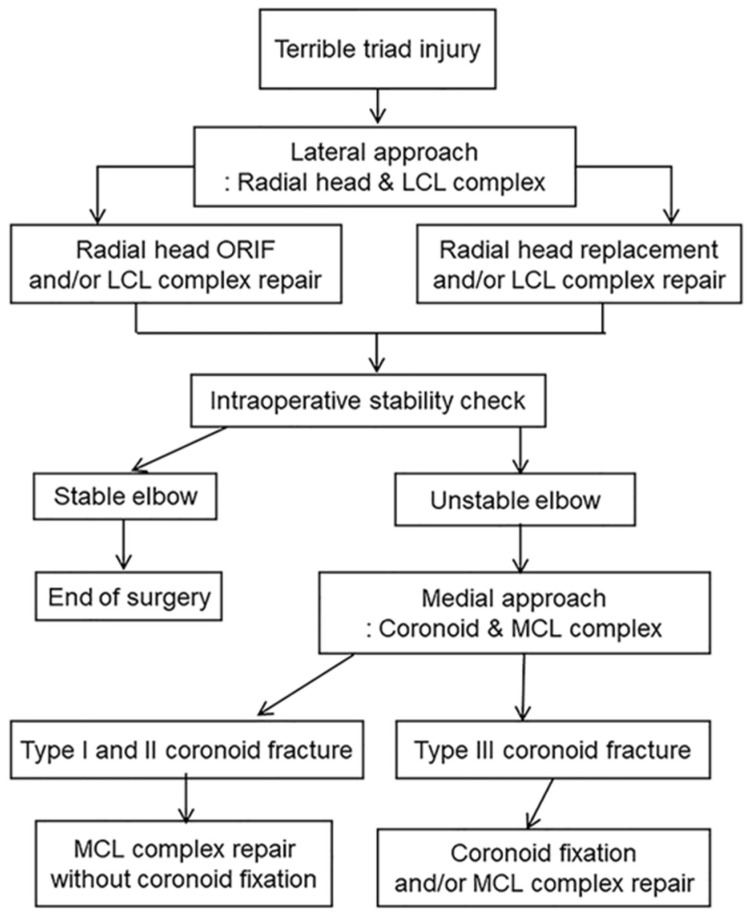
Standardized surgical protocol for terrible triad injury.

**Figure 2 jcm-09-03500-f002:**
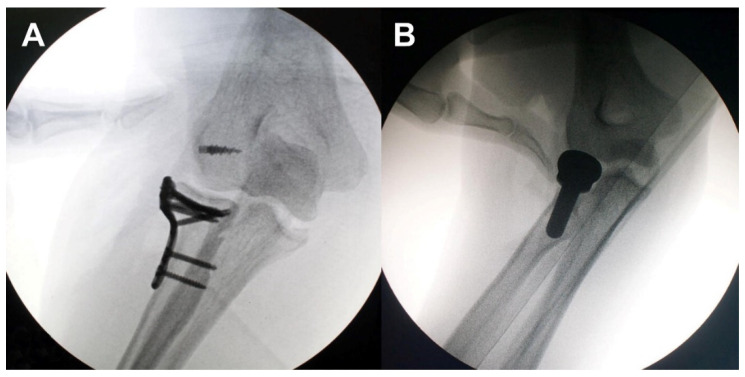
Assessment of elbow stability using valgus stress test after restoration of lateral structures. Stable (**A**) and unstable (**B**).

**Figure 3 jcm-09-03500-f003:**
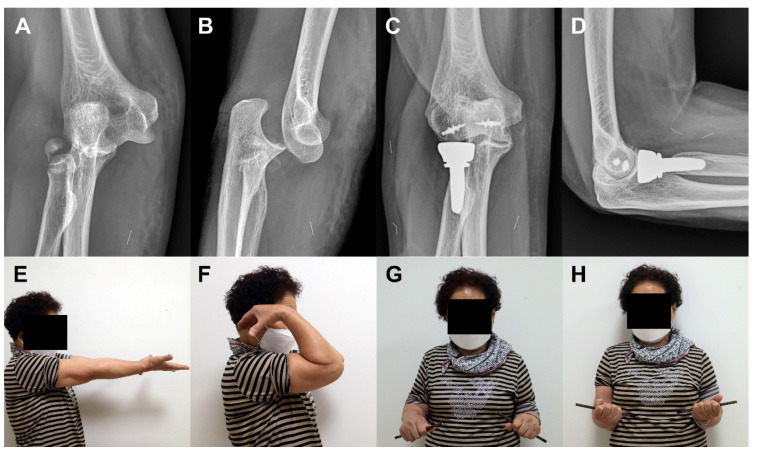
A 68-years old woman (Case 1) with terrible triad injury. Plain radiographs at the time of initial trauma reveal radial head and neck comminuted fracture, coronoid tip fracture, and elbow dislocation (**A**,**B**). Plain radiographs and clinical photos at 47 months after surgery reveal excellent radiographic and clinical outcomes (**C**–**H**).

**Figure 4 jcm-09-03500-f004:**
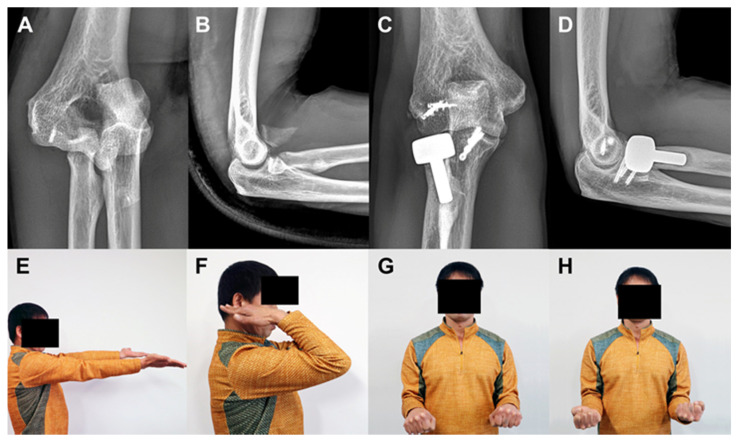
A 49-years old man (Case 11) with terrible triad injury. Plain radiographs at the time of initial trauma reveal radial head comminuted fracture, basal fracture of coronoid process, and elbow dislocation (**A**,**B**). Plain radiographs and clinical photos at 71 months after surgery reveal excellent clinical outcomes in spite of periprosthetic lucency (**C**–**H**).

**Table 1 jcm-09-03500-t001:** Demographic data of 24 patients with terrible triad injury.

Case	Sex	Age (Year)	Affected Side	Injury Mechanism	Radial Head Fx. Mason Classification	Coronoid Fx. Regan-Morrey Classification	Coronoid Fx. O’Driscoll Classification	Combined Injury	Time from Injury to Operation (Day)	Follow-Up (Mo)
1	F	68	R	Slip down	III	I	Tip 1		1	47
2	M	44	R	Fall from a height	III	II	Anteromedial 2		4	40
3	F	67	R	Slip down	III	I	Anteromedial 2	Triceps rupture	4	31
4	M	55	L	Slip down	III	II	Tip 2		1	31
5	M	39	L	Fall from a height	III	II	Tip 2	Pelvic bone Fx.	6	25
6	F	41	L	Fall down	III	I	Tip 1		2	24
7	F	67	R	Slip down	III	II	Anteromedial 2		7	163
8	M	43	R	Fall down	III	I	Tip 2		3	103
9	F	51	L	Slip down	III	II	Tip 2	Olecranon split Fx.	7	94
10	F	38	L	Sports injury	III	II	Anteromedial 1		6	75
11	M	49	R	Fall down	III	III	Basal 1		4	71
12	F	42	R	Slip down	III	I	Tip 1		1	71
13	M	42	L	Fall from a height	III	II	Tip 1		6	70
14	M	19	R	Fall from a height	III	II	Tip 2		4	36
15	F	77	R	Motor vehicle accident	III	I	Tip 1	Distal radius Fx.	18	101
16	F	62	R	Slip down	II	II	Anteromedial 2		7	38
17	F	56	L	Slip down	II	I	Tip 1	Triceps rupture	4	36
18	M	25	L	Fall down	II	III	Basal 1		4	35
19	M	37	L	Fall from a height	III	III	Basal 2	Olecranon Fx.	7	35
20	F	58	R	Sports injury	II	II	Anteromedial 2		6	144
21	M	50	L	Slip down	III	III	Basal 1		12	126
22	M	40	R	Fall down	III	II	Anteromedial 2	Lat. condylar Fx.	2	65
23	M	24	L	Slip down	III	I	Tip 1		3	60
24	F	51	R	Slip down	I	II	Anteromedial 2		7	38

**Table 2 jcm-09-03500-t002:** Summary of the clinical outcomes and complication after operative treatment in 24 patients with terrible triad injury.

Case	Radial Head Treatment	LCLC Repair	MCLC Repair	Coronoid Fixation	PO Immobilization (Day)	Final Clinical Score		Final ROM				Complications
						MEPS	Quick DASH	Flex	Ext	Pro	Sup	
1	Replacement	O	O	X	7	100	12	120	0	80	80	Transient UN
2	Replacement	O	X	X	5	100	14	140	0	80	80	
3	Replacement	O	X	X	10	85	16	140	0	80	80	
4	Replacement	O	X	X	4	95	15	140	0	80	80	
5	Replacement	O	X	X	7	85	24	125	0	80	80	
6	Replacement	O	X	X	4	90	16	140	0	80	80	
7	Replacement	O	X	X	14	80	27	140	0	80	80	
8	Replacement	O	X	X	7	80	25	120	20	80	80	Transient UN
9	Replacement	O	O	X	14	100	12	140	0	80	80	
10	Replacement	O	X	X	3	100	12	140	10	80	80	Stiff elbow, HO, UN
11	Replacement	O	X	O	4	100	15	140	0	80	80	
12	Replacement	O	X	X	5	90	19	140	0	80	80	Transient UN
13	Replacement	O	X	X	6	95	14	140	0	80	80	
14	Replacement	O	X	X	3	90	15	140	0	80	80	
15	Replacement	O	X	X	7	85	23	140	0	80	80	
16	Fixation	O	X	X	6	100	13	140	0	80	80	
17	Fixation	O	O	X	10	90	15	130	10	80	80	Olecranon Fx.
18	Fixation	O	O	O	8	100	15	130	5	70	80	
19	Fixation	O	X	O	7	70	28	125	30	50	70	Stiff elbow, HO
20	Fixation	X	X	X	7	80	26	140	0	80	80	
21	Fixation	X	X	O	7	90	14	130	0	70	80	
22	Fixation	O	O	X	14	95	18	140	5	50	80	
23	Fixation	O	X	X	7	95	16	140	0	80	80	
24	X	O	X	X	5	100	12	140	5	80	80	

**Table 3 jcm-09-03500-t003:** Correlations between variables and final clinical outcomes.

	*p* Value	
	MEPS	Quick DASH Score
Age	0.774	0.672
Sex	1.000	0.514
Affected side	0.820	0.531
Time from injury to operation	0.522	0.802
Radial head (fixation vs. replacement)	0.907	0.907
Coronoid (fixation vs. no fixation)	0.911	0.852
Duration of immobilization	0.177	0.147

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
