# Peer review of "Does the Coronoid Always Need to Be Fixed in Terrible Triad Injuries of the Elbow? Mid-Term Postoperative Outcomes Following a Standardized Protocol"

_jcm, 2020, doi:10.3390/jcm9113500_

Round 1

Reviewer 1 Report

This is a wll conducted observational retrospective study with a seemingly consitent protocoll. 

For better reproducibility the authors should provide in addition to the Regan Morrey Classification the CT based 0’Driscol classification to better delineat the non operative Types from the operative types.

Line 69: Presents a Table with 17 Patients instead of 24 patients

Line 70: The authors state that the populations observed consisted of 24, however the state that they had 20 men and 2 female patients. Had two patient a bilateral terrible triad dislocation. 

Please clarify this inconsistencies. 

Author Response

<Response to Reviewer 1 Comments>

This is a wll conducted observational retrospective study with a seemingly consitent protocoll.

For better reproducibility the authors should provide in addition to the Regan Morrey Classification the CT based 0’Driscol classification to better delineat the non operative Types from the operative types.

-> Thanks for your comment. As your comment, we added O’Driscoll classification in Table 1 & method section. (Line 77~80, page 5)

Line 69: Presents a Table with 17 Patients instead of 24 patients

->Thanks for your comment. We are sorry for our mistake. We revised Table 1.

Line 70: The authors state that the populations observed consisted of 24, however the state that they had 20 men and 2 female patients. Had two patient a bilateral terrible triad dislocation.

->Thanks for your comment. We ar very for our mistake. We revised to “there were 12 men and 12 women”. (Line 73, Page 5)

Reviewer 2 Report

General comment:

The authors present a nice study on the treatment of terrible triad injuries. The presented protocol for the treatment of these severe injuries seems to be reasonable from a clinical and biomechanical point of view and has already been described in the literature. The clinical results of the present study are extremely good and are significantly better in comparison to the literature. It would therefore be useful if the authors could present one or better 2 cases with clinical and radiological outcome.

Furthermore, some important information is still missing, which should be supplemented in the course of a revision.

Specific commentary:

abstract:

Line 17: Was not reconstructed in all patients with LCL?

Introduction:

Line 28 The Terrible Triad injury is a "common" rather than an "uncommon" injury

Lines 48 and 56 Although the Regan & Morrey classification is common, it is now obsolete as it does not take into account the approach of the anterior portion of the MCL to the anterolateral facet of the coronoid. Therefore, current studies tend to use the O'Driscoll classification. Has this issue been considered?

Methods:

Line 66 and Table 1 There is a discrepancy here. Allegedly 24 patients were included, but only 17 patients are listed in Table 1.

Line 70 Were there now 17, 22 or 24 cases?

Row 82 How was the stability investigated? One case with stable elbow and 1 case with unstable elbow would be desirable.

Line 83 Was the MCL approach used for Type II injuries?

Line 86 Which prosthesis was used?

Line 88 Which bone anchors were used?

Line 100 Did the patients receive ossification prophylaxis?

Who examined the patients?

Results:

The results, especially in patients with radial head prosthesis, are unusually good, even in comparison with the literature. It would be useful to show 1-2 patient cases with documentation of the postoperative clinical and radiological outcome.

Discussion:

ok

Author Response

<Response to Reviewer 2 Comments>

General comment:

The authors present a nice study on the treatment of terrible triad injuries. The presented protocol for the treatment of these severe injuries seems to be reasonable from a clinical and biomechanical point of view and has already been described in the literature. The clinical results of the present study are extremely good and are significantly better in comparison to the literature. It would therefore be useful if the authors could present one or better 2 cases with clinical and radiological outcome.

->Thanks for your comment. According to your suggestion, we added Figure 3 of case with clinical and radiological outcome.

Furthermore, some important information is still missing, which should be supplemented in the course of a revision.

Specific commentary:

abstract:

Line 17: Was not reconstructed in all patients with LCL?

->Thanks for your comment. Surprisingly, we did not find LCL tear in 2 cases in spite of comminuted radial head fracture.

Introduction:

Line 28 The Terrible Triad injury is a "common" rather than an "uncommon" injury

->Thanks for your comment. We revised to “common”. (Line 27~28, Page 1)

Lines 48 and 56 Although the Regan & Morrey classification is common, it is now obsolete as it does not take into account the approach of the anterior portion of the MCL to the anterolateral facet of the coronoid. Therefore, current studies tend to use the O'Driscoll classification. Has this issue been considered?

->Thanks for your comment. As your comment, we added O’Driscoll classification in Table 1 & method section. (Line 77~80, Page 99)

Methods:

Line 66 and Table 1 There is a discrepancy here. Allegedly 24 patients were included, but only 17 patients are listed in Table 1.

->Thanks for your comment. We are sorry for our mistake. We revised Table 1.

Line 70 Were there now 17, 22 or 24 cases?

->Thanks for your comment. We are sorry for our mistake. We revised the sentence.

Row 82 How was the stability investigated? One case with stable elbow and 1 case with unstable elbow would be desirable.

->Thanks for your comment. We did valgus stress test under the image intensifier after complete reconstruction of lateral structures. According to your suggestion, we revised this sentence and added Figure 2 as stable or unstable elbows.

Line 83 Was the MCL approach used for Type II injuries?

->Thanks for your comment. Yes, we did. “If instability remained, the coronoid process and/or MCL complex were fixed through a medial approach.”

Line 86 Which prosthesis was used?

->Thanks for your comment. We added name of prosthesis. (Line 91~93, Page 5)

Line 88 Which bone anchors were used?

->Thanks for your comment. We added name of suture anchor. (Line 95~96, Page 5)

Line 100 Did the patients receive ossification prophylaxis?

->Thanks for your comment. Yes, we did not use anything for prophylaxis of heterotopic ossification.

Who examined the patients?

->Thanks for your comment. All radiographic outcomes were examined by a senior surgeon (corresponding author).

Results:

The results, especially in patients with radial head prosthesis, are unusually good, even in comparison with the literature. It would be useful to show 1-2 patient cases with documentation of the postoperative clinical and radiological outcome.

->Thanks for your comment. According to your suggestion, we added Figure 3 of case with clinical and radiological outcome.

Round 2

Reviewer 2 Report

I agree with the authors that the patient in Figure No.3 had a good clinical outcome measured by the severity of the injury. However, the flexion is 120° maximum, probably more like 110°. However, in the overview of the results, there is only one patient with a flexion of 120°, but this patient has an extension of 20°. So the flexion of this patient was probably not measured exactly.

As mentioned in my first review, the results presented are unusually good and do not correspond to reality in a literature comparison. Therefore, the authors should critically review all patients again and indicate the correct extent of movement etc.

Furthermore, it would be nice if the authors could show a case study with medial treatment and treatment of a coronoid fracture.

Author Response

<Response to Reviewer 2 Comments>

I agree with the authors that the patient in Figure No.3 had a good clinical outcome measured by the severity of the injury. However, the flexion is 120° maximum, probably more like 110°. However, in the overview of the results, there is only one patient with a flexion of 120°, but this patient has an extension of 20°. So the flexion of this patient was probably not measured exactly.

->Thanks for your comment. We are sorry and do not deny the errors of measurement of ROMs. In this study, however, all clinical outcomes including the MEPS, Quick-DASH score, all ROMs were evaluated by an independent research coordinator. In this case, the discrepancy may cause the mistake during taking photo. Anyway, we revised ROMs in this case.

As mentioned in my first review, the results presented are unusually good and do not correspond to reality in a literature comparison. Therefore, the authors should critically review all patients again and indicate the correct extent of movement etc.

->Thanks for your comment. This study revealed that operative treatment based on our standardized protocol for terrible triad injuries yielded satisfactory mid-term clinical and radiographic outcomes without any recurrent instability. These results suggest that Type I and II coronoid fractures in terrible triad injuries do not need to be fixed if the radial head and ligamentous complex are completely reconstructed. In the other words, this study demonstrated that minimal reconstruction to obtain the stability and early rehabilitation can produce excellent outcomes, although terrible triad injures have traditionally reported unsatisfactory outcomes after treatment. We believe that these are why the results from our study are superior compared to those of previous studies.

We do not deny the errors of measurement of ROMs. In this study, however, all clinical outcomes including the MEPS, Quick-DASH score, all ROMs were evaluated by an independent research coordinator. We tried to evaluate objectively for each patient.

Furthermore, it would be nice if the authors could show a case study with medial treatment and treatment of a coronoid fracture.

->Thanks for your comment. According to your suggestion, we added Figure 4 of case with coronoid fixation.